# Effect of N-Acetyl Cysteine as an Adjuvant Treatment in Alzheimer’s Disease

**DOI:** 10.3390/brainsci15020164

**Published:** 2025-02-07

**Authors:** Sarah Monserrat Lomelí Martínez, Fermín Paul Pacheco Moisés, Oscar Kurt Bitzer-Quintero, Javier Ramírez-Jirano, Daniela L. C. Delgado-Lara, Irán Cortés Trujillo, Juan Heriberto Torres Jasso, Joel Salazar-Flores, Erandis Dheni Torres-Sánchez

**Affiliations:** 1Department of Medical and Life Sciences, Cienega University Center, University of Guadalajara, Ocotlan 47820, Jalisco, Mexico; sarah.lomeli@academicos.udg.mx (S.M.L.M.); iran.cortes@academicos.udg.mx (I.C.T.); joel.salazar@academicos.udg.mx (J.S.-F.); 2Periodontics Specialty Program, Department of Integrated Dentistry Clinics, University Center for Health Sciences, University of Guadalajara, Guadalajara 44340, Jalisco, Mexico; 3Institute of Research in Dentistry, Department of Integral Dental Clinics, University Center for Health Sciences, University of Guadalajara, Guadalajara 44340, Jalisco, Mexico; 4Public Health, Department of Wellbeing and Sustainable Development, Northern University Center, University of Guadalajara, Colotlán 46200, Jalisco, Mexico; 5Chemistry Department, University Center of Exact Sciences and Engineering, University of Guadalajara, Guadalajara 44430, Jalisco, Mexico; fermin.pacheco@academicos.udg.mx; 6Neurosciences Division, Western Biomedical Research Center, Mexican Social Security Institute, IMSS, Guadalajara 44340, Jalisco, Mexico; neuronim26@yahoo.com (O.K.B.-Q.); ramirez_jirano@hotmail.com (J.R.-J.); 7Departamento Académico de Formación Universitaria, Ciencias de la Salud, Universidad Autónoma de Guadalajara, Zapopan 45129, Jalisco, Mexico; daniela.delgadolara@gmail.com; 8Department of Biological Sciences, University Center of the Coast, University of Guadalajara, Puerto Vallarta 48280, Jalisco, Mexico; heriberto.torres@academicos.udg.mx

**Keywords:** N-acetyl-cysteine, Alzheimer’s disease, oxidative stress, mitochondrial dysfunction, Aβ peptides, hyperphosphorylation of tau

## Abstract

Oxidative stress levels are exacerbated in Alzheimer’s disease (AD). This phenomenon feeds back into the overactivation of oxidase enzymes, mitochondrial dysfunction, and the formation of advanced glycation end-products (AGEs), with the stimulation of their receptors (RAGE). These factors stimulate Aβ peptide aggregation and tau hyperphosphorylation through multiple pathways, which are addressed in this paper. The aim of this study was to evaluate the regulatory effect of N-acetyl cysteine (NAC) on oxidant/antioxidant balance as an adjuvant treatment in patients with AD. The results obtained showed that NAC supplementation produced improved cognitive performance, decreased levels of oxidative stress markers, lowered activities of oxidase enzymes, increased antioxidant responses, and attenuated inflammatory and apoptotic markers. Moreover, NAC reversed mitochondrial dysfunction, lowered AGEs-RAGE formation, attenuated Aβ peptide oligomerization, and reduced phosphorylation of tau, thereby halting the formation of neurofibrillary tangles and the progression of AD.

## 1. Introduction

Alzheimer’s disease (AD) and other neurological pathologies are associated with increased levels of oxidative stress. This phenomenon involves triggering the generation of oxidative species, particularly those derived from oxygen and nitrogen, i.e., reactive oxygen species (ROS) and reactive nitrogen species (RNS), respectively. The brain and central nervous system (CNS) are susceptible targets for attack by ROS and RNS, due to several factors: high concentrations of mitochondria, low levels of antioxidants, presence of polyunsaturated fatty acids (targets for oxidation) in the basal tissue, high rates of oxygen consumption, presence of iron-dependent metabolic pathways, and the generation of nitric oxide [1,2,3]. In the literature, ROS and RNS are categorized as free radicals that are highly reactive with nucleic acids, proteins, and polyunsaturated lipids. The major types of ROS are superoxide anion (O_2_^.^), hydroxyl radical (OH^.^), and hydrogen peroxide (H_2_O_2_). These ROS are generated at the mitochondrial level during oxidative phosphorylation by the activation of oxidase enzymes involved in signaling processes and by neurochemical reactions regulated by Cu(I) and Fe(II) ions in the Fenton and Haber-Weiss reactions. The most common RNS are nitric oxide (NO), peroxynitrite (ONOO-), nitrites (NO_2_^.^), and nitrates (NO_3_^.^). In the CNS, NO release is predominant at the neuronal level, and it is mediated by neuronal nitric oxide synthase enzymes (nNOS) and mitochondrial nitric oxide synthase (mtNOS). Once released, the NO radical reacts with O_2_^.^ to generate ONOO-. Physiologically, the NO molecule has a neuro-modulatory role in cognitive functions involved in memory processing, potentiation, and synaptic plasticity. However, on depletion of antioxidant capacity, NO exacerbates neuroinflammation, which may aggravate AD in patients [4,5,6,7]. At the functional level, the production of ROS and RNS is maintained within the homeostatic range, thanks to the activities of enzymatic and non-enzymatic antioxidants, which suppress oxidation. In patients with early-to-moderate AD, an attenuating effect on the progression of this pathology was reported as a result of prescription and use of antioxidant therapy [8]. Basically, antioxidants increase the rate of removal of ROS/RNS and decrease their formation through the following mechanisms: (a) electron transfer to the free radical; (b) activation of antioxidant enzymes such as superoxide dismutase (SOD), catalase (CAT) and glutathione peroxidase (GPx); (c) activation of non-antioxidant enzymes which induce endogenous synthesis of antioxidant molecules; (d) inhibition of enzymes that generate ROS/RNS; (e) processes that induce the chelation of metal ions, thereby reducing the formation of ROS/RNS; (f) absorption of ultraviolet radiation; and (g) attenuation of reactions involving molecular oxygen (O_2_) [9,10]. These mechanisms are summarized in Figure 1.

Antioxidants are classified as preventive, radical scavenging, reparative, and adaptive antioxidants. An antioxidant that has gained prominence in recent years is N-acetylcysteine (NAC), which is a precursor of reduced glutathione (GSH) with antioxidant capacity. NAC attenuates the release of NO_2_ and ROS, facilitates the hydrolysis of enzymes/proteins with thiol groups, and stimulates the release of albumin, which is classified as an antioxidant [9,10]. Some previous studies have evaluated the antioxidant effect of NAC in NAC-supplemented murine models of AD. Decreases in the formation of senile beta-amyloid (Aβ) plaques and lower production of lipid peroxidation in hippocampal and prefrontal tissues were observed. Moreover, similar results were observed in AD patients given NAC supplementation. In these patients, NAC induced low levels of oxidative stress in the hippocampus and increased intracellular GSH synthesis. Moreover, in other studies on AD patients, it was reported that NAC attenuated the progression of dementia and exerted a neuroprotective effect, although the latter effect was not conclusive [11,12,13]. Therefore, the present study was focused on the evaluation of the effect of NAC as an adjuvant treatment on various pathways that regulate the oxidant/antioxidant balance in patients with AD.

## 2. Oxidative Stress in Patients with AD

AD is related to an oxidant/antioxidant imbalance. This imbalance in favor of oxidation seems to act as a bridge in the formation of senile plaques due to the accumulation of Aβ and neurofibrillary tangles [6,9]. However, it is not clear whether oxidative stress triggers the formation of senile Aβ plaques and neurofibrillary tangles or whether senile Aβ plaques and neurofibrillary tangles increase oxidative stress [12,14,15]. Physiologically, neurons exhibit the phenomenon of hormesis, which describes free radicals as inducers of neuroprotection: increasing the release of ROS stimulates NF-kβ transcription factors related to the expressions of Mn-SOD genes with antioxidant capacity. On the other hand, it has been reported that NO may also be neuroprotective by regulating signal transduction through nitrosylation. The problem arises when the depletion of antioxidant capacity tips the balance towards oxidation, as seen in patients with AD [2,6,16,17].

The main factors that contribute to the accentuated oxidative stage in this pathology are described below:

Firstly, as depicted in Figure 2, patients with AD show overactivation of different types of oxidase enzymes, i.e., NADPH oxidases (NOX), cytochrome c oxidase (Coc), and monoamine oxidase B (MAO-B) [6,15].

The NOX enzymes are involved in phosphorylation processes and in the regulation of transcription factors that modulate the vasculature. There are several types of NOX, depending on their isoforms, but for this review, NOX2 and NOX4, which are located mainly in the cortical area and frontal lobes in AD patients, stand out. The isoform NOX2 is found in microglial cells with macrophage function in the CNS. Thus, it functions in the modulation of neuronal apoptosis. For activation, NOX2 requires electron transfer with an increase in O_2_^.^ release. The NOX4 isoform is inducible, and it is found in the cerebral vascular endothelium and astrocytes. Increases in ROS correlate with levels of NOX4 mRNA. When NOX4 is over-activated, H_2_O_2_ production is induced, thereby further exacerbating the oxidative state in AD. Finally, the activation of NOX2 and NOX4 alters the pH of the medium, which may further favor the formation of senile plaques from Aβ aggregates [18,19].

Inside the mitochondrion, the Coc complex transfers electrons with lower energy potential to oxygen, resulting in the release of water. This allows for the subsequent transport of protons to the intermembrane space, where they are fed to ATP synthase for the generation of ATP. However, alterations in the enzymatic subunits that assemble Coc have been reported in AD patients, resulting in the destabilization and inhibition of the enzyme complex. In murine models, it was found that increased ROS peroxidizes the mitochondrial inner membrane, thereby destabilizing Coc. It is important to note that the depletion of Coc is associated with neurodegeneration and amyloidogenesis [20].

Similarly, MAO-B is a mitochondrial enzyme that catalyzes oxidative deamination reactions of monoamine-type neurotransmitters. However, overactivity of this enzyme is associated with increased levels of H_2_O_2_, aldehydes, and ammonia. Interestingly, MAO-B promotes the cleavage of amyloid precursor protein (APP), which contributes to the formation of senile plaques and neurofibrillary tangles, with the consequence of neurodegeneration [21,22].

Secondly, the phenomenon known as mitochondrial dysfunction also occurs [6,15]. A high energy consumption is required in the CNS, which is provided by the production of ATP by mitochondrial oxidative phosphorylation at four enzyme complexes in a reaction catalyzed by ATP synthase. Between the process regulated by the third enzyme complex and fourth enzyme complex (Coc), O_2_^.^ is released, which should be reduced to water. However, in some situations, this reduction process fails, resulting in the release of the highly reactive O_2_^.^ radical. This type of ROS alters the mitochondrial DNA, resulting in deleterious changes in important proteins that form the enzyme complexes mentioned above, thereby generating the phenomenon known as mitochondrial dysfunction, which is responsible for increased synthesis of free radicals. At the early stages of AD, mitochondrial dysfunction has an impact on the polymerization of Aβ peptides. In addition, the increases in levels of free radicals induce nitration, carbonylation, and peroxidation of important metabolic enzymes and biomolecules such as ATP synthase and phospholipids. These reactions alter mitochondrial stability in the hippocampus and parietal cortex in patients with mild cognitive impairment. These aforestated changes lead to the release of Fe(III) ions, which further contribute to mitochondrial dysfunction and ultimately to neuronal death [2,15,23].

Thirdly, it has been observed that the Met35 residue of the Aβ peptide increases the levels of free radicals [6,15]. The Aβ peptide is released by cleavage by APP, and the main peptides released are Aβ40 and Aβ42, which have 39 to 43 amino acids. It is important to note that amino acid in position 35 is methionine (Met35), which has sulfur in its radical chain. Chemically, methionine is susceptible to redox reactions, with oxidation requiring the loss of a pair of electrons to form methionine sulfate, i.e., oxidized form (MetSO), which is then reduced by the enzyme methionine sulfoxide reductase to convert it back to methionine. These reductase enzymes have antioxidant roles in that they act as ROS chelators. However, in AD, there are decreased activities of reductases, leading to increases in the release of MetSO and ROS, thereby worsening the oxidative state. Indeed, decreases in activities of methionine sulfoxide reductases lead to increases in oligomerization of Aβ associated with AD pathogenesis. In vitro studies in which the amino acid methionine at position 35 was replaced with other amino acids such as isoleucine, leucine, lysine, and tyrosine have resulted in inhibition of the aggregation of Aβ peptides [24,25].

Fourthly, the Aβ peptide has binding affinity for metal ions such as Zinc (Zn), copper (Cu) and iron (Fe), resulting in formation of the metal complexes Zn(II)-Aβ, Cu(I/II)-Aβ and Fe(II/III)-Aβ, respectively. The polymerization of senile plaques is stabilized by the binding of Aβ to these metals. Similarly, the interaction of the Aβ-metal ion favors the reduction of Cu(I/II) and Fe(II/III), which increases their reactivity with oxygen, leading to release of O_2_^.^. It has also been reported that reduced Fe(II/III) reacts easily with H_2_O_2_, thereby generating OH^.^ Radicals via the Fenton reaction. Thus, the increases in levels of ROS favor the aggregation of Aβ peptides, which prolong the feedback of the oxidative cycle, as described in the hippocampus of AD patients [6,15,24].

Fifthly, collateral damage that occurs as a consequence of increased levels of free radicals and mitochondrial dysfunction in patients with AD is compromised metabolism of glucose and calcium. The synergy between elevated glucose levels and free radicals triggers the formation of advanced glycation products (AGEs) and the stimulation of their receptors (RAGE). The AGEs modify soluble neurofilament proteins into insoluble aggregates, a process which occurs when RAGE binds to Aβ peptides, thereby inducing the release of ROS and, as a consequence, favoring the oligomerization of these peptides. It has been reported that RAGE-Aβ complexes may be detected in early-onset AD [15,26]. Reports have shown that there are high levels of AGEs in neurons, astrocytes, and glia of AD patients. The higher the concentrations of AGEs, the higher the levels of oxidative stress, neuroinflammation, and polymerization of Aβ peptides as well as Tau aggregates. However, the directionality of the relationship between AGEs and neurodegeneration in patients with AD has not been precisely described, although, so far, everything points to a bidirectional or cyclical relationship [27].

To sum up, the increases in levels of ROS through the five pathways described above increase in parallel, the release of RNS, and nitro oxidative stress has an impact on tau phosphorylation, which results in neurodegeneration and apoptosis [28]. Therefore, the aim of therapy is to shore up the antioxidant capacity in patients with AD so as to counteract the oxidative damage that leads to neurodegeneration.

## 3. NAC

### 3.1. General Characteristics of NAC

NAC (C_5_H_9_NO_3_S) is a compound synthesized through the linkage of cysteine to an acetyl group. The compound, which has a molecular weight of 163.2 g/mol, was developed in 1960, and it is characterized by the presence of a sulfhydryl group (Figure 3) [29,30].

NAC is highly soluble in water, and it is rapidly absorbed via anion exchange. The bioavailability of NAC ranges from 4% to 10%, with 66% to 97% binding to albumin proteins. It has a half-life (t_½_) of 5.58 h when administered intravenously (i.v.) and a t_½_ of 6.25 h when given orally. The hepatic metabolism of NAC involves deacetylation and, in some cases, oxidation to a disulfide known as N,N′ diacetyl cysteine. The main products released from the metabolism of NAC are cysteine, cystine, inorganic sulfate, N,N′ diacetyl cysteine, and GSH. The synthesis of GSH is carried out in two steps. The first step is catalyzed by the enzyme γ-glutamylcysteine synthetase, and it involves the condensation of glutamate and cysteine to form γ-glutamylcysteine in the presence of ATP. In the second step, the enzyme GSH synthetase converts γ-glutamylcysteine to GSH. The clearance of NAC is mainly renal, although hepatic clearance occurs to a lesser extent (13–38% via the urine and 3% via the feces). Therapeutically, paracetamol toxicity is counteracted with NAC prescription in patients with cystic fibrosis, in conditions with inspissated mucous secretions as pneumonia and bronchitis, and for patients with chronic obstructive pulmonary disease (COPD), as it does not increase the mucus volume, and for patients with nephropathy. It is also used as an anti-inflammatory agent, antioxidant, heavy metal chelator, antiviral drug, vascular permeability regulator, regulator of ATP and NO synthesis, as well as in CNS disorders, cardiovascular diseases, and retinitis pigmentosa. The adverse effects reported following the administration of NAC are gastrointestinal (upset stomach, Abdominal distension, diarrhea, nausea, vomiting), and skin and systemic alterations. In more severe cases, anaphylactic shock may occur. The adverse effects of NAC overdose are hemolysis, thrombocytopenia, acute renal failure, and death. It is important to emphasize that the frequent use of NAC in patients with AD may have some adverse effects, so it is recommended that NAC be prescribed orally to mitigate anaphylactic reactions. In addition, as continuous use of NAC may have an impact on gastric changes, the patient should be assessed for pre-existing gastrointestinal ulcers or varicose veins. Liver and renal function tests should be evaluated prior to NAC consumption and periodically to ensure that the dose of NAC is adequate and does not have repercussions in lesions [9,11,29,30,31,32,33,34,35].

In the use of NAC as an antioxidant treatment, one issue that needs to be resolved is the conflicting data on whether or not it crosses the blood-brain barrier (BBB). In this regard, it has been reported that NAC crosses the BBB when the membrane is breached by increased oxidative stress [29]. On the other hand, it has been shown that in order to cross the BBB, there is need for an active transport mechanism regulated by a carrier, and it is also necessary for NAC to undergo deacetylation, although it is important to point out that its transport across the BBB depends on the route of administration. For example, it has been reported that intra-arterial and intravenous administration of 14C-NAC resulted in the highest levels of BBB permeability when compared to other routes of administration. The BBB permeability is higher for NAC amide and NAC ethyl ester than for the simple form of NAC [36,37].

### 3.2. Antioxidant Properties of NAC

The antioxidant capacity of NAC lies in its sulfhydryl (SH) group, which is a nucleophile that donates one or two electrons. Likewise, the amino group of NAC acts as an electron acceptor or donor in the reduction of ROS and RNS. In particular, NAC has a high chemical affinity for the reduction of OH^.^, NO_2_^.^, and nitroxyl (HNO) radicals. On the other hand, the SH group of NAC and the formation of GSH favor the scavenging of transition metal ions such as Cu(II) and Fe(III), thereby facilitating their stabilization. In addition, NAC favors the cleavage of extracellular proteins with disulfide bridges via an SN2 reaction mechanism where the SH of NAC attacks the central sulfur of the protein disulfide bridge, thereby releasing a thiol, which increases their antioxidant capacity. This modifies the structures and even the functions of proteins, and it is often beneficial when it comes to changes in tumor necrosis factor alpha (TNF-α) receptors. Due to its anti-inflammatory potential, NAC inhibits the nuclear factor kappa chain enhancer of activated B cells (NF-kB), thereby limiting its translocation and proinflammatory activation by decreasing the release of interleukins (IL) type 1β and type 6 [32,33].

The synthesis of the antioxidant GSH requires cysteine, glutamate, and glycine. Since this synthesis depends on the concentration of cysteine, the administration of NAC is an excellent way of guaranteeing adequate GSH concentrations. This antioxidant is one of the main lines of defense in counteracting oxidation by free radicals, particularly H_2_O_2_, hydroperoxides, and oxoaldehydes. Decreased GSH levels have been implicated in the aging process and also in the development of neurodegenerative diseases [32,33].

It has been reported that cysteine has a greater antioxidant capacity than GSH, and in turn, GSH is a better antioxidant than NAC, a phenomenon related to the reduction exerted by each compound as a function of the alkalinity of the SH group. Therefore, NAC by itself is classified as a weak antioxidant, and its main contribution lies in the release of cysteine and the formation of GSH. It should be noted that, to a lesser extent, the thiol groups of cysteine, GSH, and NAC act as prooxidants that generate OH^.^ and thiol radicals, and when they interact with Cu(II) or vitamin B12, they also act as prooxidants. However, so far, this increase in prooxidant potential has been reported in vitro only, and not in vivo, as shown in Figure 4 [29,38].

In summary, the mechanism of action of NAC involves (a) restoration of cellular antioxidant potential by increasing GSH levels, (b) elimination of free radicals, and (c) inhibition of proinflammatory factors and enzymes [30].

### 3.3. Properties of NAC in CNS and AD

It has been reported that in the CNS, NAC supplementation resulted in neurogenic and neuroprotective properties, which are also related to increased dopamine release from presynaptic neurons to the inter-synaptic space. Unfortunately, NAC supplementation leads to the release of glutamate both from the presynaptic neuron and glial cells since the entry of NAC into these cells is via antiporter transport with glutamate into the inter-synaptic space. The binding of dopamine and glutamate to their respective receptors (dopaminergic, AMPA, and NMDA) physiologically increases levels of ROS and RNS in the postsynaptic neuron. However, supplementation with NAC increases GSH levels, thereby counteracting oxidation. At the mitochondrial level, NAC supplementation not only modulates the redox state but also lowers Ca^2+^ ion levels, thereby decreasing the apoptotic response [32,33]. It should be noted that in clinical trials focused on neurodegenerative diseases, the oral supplementation doses of NAC range from 600 to 1250 mg/day, with a duration ranging from 4 weeks to 6 months. However, in patients with a presumptive diagnosis of AD, NAC has been prescribed for up to 6 months at doses ranging from 50 to 600 mg/day [38,39,40,41,42]. In patients with mild cognitive impairment who consumed NAC at a dose of 600 mg/day for 6 months, improvements in the dementia assessment scale and in the preservation of cognitive function have been reported. Similar improvements in cognitive function were obtained in studies on murine models, although there were no decreases in Aβ plaque formation [11,32,33,43].

Table 1 below details the studies that evaluated the effect of NAC in clinical trials, animal models, and in vitro studies, either by CNS alterations or by AD.

Unfortunately, there are few clinical trials that evaluated the effect of NAC in patients with cognitive impairment as well as its effect on markers of oxidative, inflammatory, and apoptotic stresses. In Table 1, we report three trials. Additionally, Mecocci and Polidori reported other studies registered in the Clinicaltrials.gov platform (NCT01370954, NCT00903695) where they gave NAC supplementation to several groups of patients with memory loss or probable AD diagnosis. The results showed that NAC, in conjunction with other nutraceuticals, improved their responses on the MMSE scale and their cognitive performance [53]. On the other hand, McCaddon and Davies report three clinical cases of patients with a probable diagnosis of AD to evaluate the cognitive response to providing 600 mg of NAC along with other supplements where they have seen improvement in the behavior of patients, in their cognitive response, and improvement in their mood [38]. In addition, supplementation of 24 older adults with NAC in combination with glycine for 16 weeks showed improvement in markers of inflammation and oxidative stress, as well as improved performance on tests of physical function [54]. These results are similar to those reported in the three trials described in Table 1 of this paper, which showed improvement in cognitive performance tests, behavior, and well-being of patients given NAC supplementation. It is important to note that only the work of Adair (2001) evaluated oxidative status before and after NAC intervention, and the results showed no differences in the levels of glutathione peroxidase, GSH and thiobarbiturate reactive substances (TBARS), which may be related to the low prescribed dose of 50 mg/kg per day [39].

From the studies described in Table 1 for various murine models of AD, NAC supplementation produced cognitive improvement, and some studies even indicated that cognitive impairment could be reversed. However, More (2018) and Darbandi (2023) reported no significant differences between the group supplemented with the NAC + AD model with respect to the AD model [47,50].

Five of the eight papers reported in Table 1 on the murine models of AD evaluated the effect of NAC supplementation on oxidative status, out of which only four reported decreases in levels of MDA, nitrite and oxidized glutathione, along with increases in antioxidant capacity due to SOD, CAT, FRAP, GST, GPx and GSH/GSSG ratio [12,44,47,49].

In addition, one of the eight murine models of AD (Table 1) reported that NAC supplementation attenuated the levels of inflammatory and apoptotic markers such as TNF-α and IL-6, while microglia activation was decreased [49].

In three of the studies described in Table 1, Aβ peptide aggregation and Tau phosphorylation were evaluated. In all the studies, it was observed that NAC supplementation in murine models of AD resulted in lower Aβ peptide aggregation and decreased Tau phosphorylation [12,46,48,49].

From the three in vitro models reported in Table 1, we observed restorations in antioxidant values of GSH and MTT and decreased oxidative status with low levels of HNE, CML, and OH-1, attenuating mitochondrial dysfunction. Similarly, NAC down-preserved Aβ-peptide levels and Tau phosphorylation [28,51,52].

Finally, the data in Table 1 show that 600 mg of NAC orally per day was the dose at which attenuation of oxidative stress and improvement in cognitive response were observed in clinical trials. On the other hand, doses ranging from 50 to 200 mg/kg i.p. per day have been shown to improve markers of inflammation, apoptosis, oxidative stress, and cognitive response in animal models. In in vitro studies, the most effective dose varies from 1 to 30 thousand μM (equivalent to 30 mM). To determine the most appropriate dose, further clinical trials in Alzheimer’s patients are needed.

In summary, Table 1 generally shows that NAC supplementation counteracted the damage caused by AD by re-establishing or inhibiting oxidative damage. This result is similar to that reported by Bavarsad Shahripour et al. (2014) and Hara (2017) [36,37], in that if NAC confers protection to reestablish oxidative stress induced by Aβ1-42 or by Tau hyperphosphorylation, this would tip the balance in favor of antioxidant response seen in the described studies like that reported by Tardiolo et al. (2018) [55]. Therefore, the roles of NAC in oxidase enzyme activity, mitochondrial dysfunction, increases in Aβ peptides in conjunction with ROS, and in the formation of AGEs products are detailed below.

## 4. NAC as an Attenuator of Oxidase Enzyme Activity

The increases in activities of NOX2 and NOX4 are linked to the induction of a higher oxidative state. In this regard, it has been reported that combined therapy with NAC and apocynin was effective in reducing NOX4 expression in renal tissue through the inhibition of MAPK-induced phosphorylation of p47phox transcription factors [56]. In addition, it has been hypothesized that NAC treatment supplies the cell with GSH, which inhibits NOX activity by decreasing p22phox gene expression and p67phox translocation to the membrane, so that NAC attenuates oxidative damage in rats, although it should be noted that the underlying mechanism is not yet fully elucidated [57].

It has been reported that NAC supplementation decreased the expression and activity of COX-2, possibly due to the decrease in NOX expression, and that at the mitochondrial level, NAC acts on NOX and even on mitochondrial NADPH-dependent reductase enzymes, thereby moving the balance towards a decrease in oxidative state [57,58,59]. Similarly, in a study using the MG63 cell line, it was observed that the addition of NAC led to inhibition of COX-2 mRNA expression induced by IL-1β. In addition, NAC prevented the nuclear translocation of NFκB, thereby stopping the inflammatory process in this cell line. It is important to emphasize that NFκB modulates inflammatory and apoptotic processes in various pathologies [60]. Another mechanism through which NAC acts is by suppressing the activity of phospholipase A2. When this enzyme is inhibited, there is a decrease in the release of arachidonic acid (which is the substrate for COX-1 and COX-2), thereby suppressing the release of proinflammatory prostaglandins [61].

Regarding the effect of NAC on MAO enzymes, it has been reported that NAC supplementation inhibited MAO activity and increased monoaminergic neurotransmission; NAC also attenuated glutamatergic transmission, thereby decreasing glutamate excitotoxicity [62,63]. The increase in monoaminergic neurotransmission implies an increase in dopamine, which may be oxidized to release dopamine-quinone and CNS toxic compounds. The advantage of supplementing with NAC is that it attenuates these oxidative products derived from dopamine [63]. Finally, it has been reported that the increase in dopamine levels in cortico-striatal regions significantly altered the CNS, while supplementation with NAC attenuated the damage that caused the increases in levels of dopamine in these regions [64].

## 5. Effect of NAC on Mitochondrial Dysfunction

Mitochondrial dysfunction involves a depletion of reducing molecules, e.g., GSH, along with low activities of SOD, CAT and GPx, leading to alterations in mitochondrial enzyme complexes involved in oxidative phosphorylation [65]. In rats with STZ-induced cognitive impairment, mitochondrial dysfunction was increased with alterations in activities of NADH dehydrogenase, succinate dehydrogenase, and cytochrome oxidase enzyme complexes. However, when murine models were given NAC supplementation at doses of 50 and 100 mg/kg, these enzyme complexes were significantly restored [66]. The same response was observed when rats with renal damage were administered NAC and apocyanin, leading to reversal of the mitochondrial dysfunction and improvement of the enzymatic activities of SOD, GSH reductase and GPx [56].

It has been observed that NAC supplementation maintained an adequate rate of mitochondrial respiration [58]. Indeed, NAC has the potential to supply the cell with GSH and cysteine, thereby reversing the prooxidant environment and restoring the activities of SOD, CAT and GPx. Moreover, NAC improves mitochondrial fission, mitophagy, and mitochondrial biogenesis, and it also acts as an electron donor through its SH group, exerts an antioxidant effect against mitochondrial ROS, and acts synergistically with antioxidant enzymes [65,67,68].

The chemical advantage of the SH group is that it regulates different protein processes important in translation, histone modification, and mRNA splicing. This may account for the decreased expressions of NOX and COX, as well as the modification of the redox balance. Likewise, this chemical group facilitates the ubiquitination of misfolded proteins, thereby facilitating the degradation of Aβ1-40 and Aβ1-42 peptides. Impairment of mitochondrial DNA, which alters the synthesis of complex IV of the respiratory chain in the temporal cortex and hippocampus, has been reported in patients with AD. However, NAC supplementation restored the mitochondrial enzyme complex IV by decreasing ROS production and increasing the activity of mitochondrial complex I in nerve cells [50,68].

## 6. Effect of NAC as an Attenuator on Aβ-Peptide Aggregation

The greater the aggregation of Aβ peptides, the greater the presence of alterations in neuronal plasticity, neuroinflammation, and cholinergic and dendritic pathways, among other consequences. Supplementation with NAC not only provides higher levels of GSH but also raises cysteine levels. In this regard, it has been reported that cysteine prevented the aggregation of Aβ1-40 and Aβ1-42 peptides, and NAC also decreased apoptosis induced by Aβ aggregation in human neurons of the cerebral cortex [68].

In the work of Alkandari et al. (2023), it was observed that the murine group supplemented with NAC-amide showed significant reductions in Aβ expression in the hippocampus and medial prefrontal cortex, although they did not match the control group [12]. These results are similar to those reported by Atlas (2021), and they demonstrate that NAC-amide supplementation, in addition to decreasing the aggregation of Aβ peptides, manages to attenuate the carbonylation of Aβ proteins and also inhibits the phosphorylation of MAPK, JNK and p38MAPK pathways, leading to reversal of Aβ1-42-induced neurotoxicity, suggesting a neuroprotective effect of NAC [45,49,69].

Finally, in vitro studies have shown the potential of NAC (1nM) to disassemble fibrils, protofibrils, oligomers and dimers of Aβ molecules by blocking continuity in their oligomerization, a phenomenon mediated by the formation of hydrogen bridges between the Aβ molecule and NAC, and by the stabilization of the neprilysin protease which is part of the catabolic equilibrium of the Aβ peptide [70,71,72].

## 7. Effect of NAC as an Attenuator of Tau Hyperphosphorylation

Hyperphosphorylation of Tau protein is observed in patients with AD, leading to the destabilization of microtubules and inhibition of axonal transport, neuronal transmission, and synaptic activity. In several studies, it has been observed that the group given NAC supplementation had decreased hyperphosphorylation of Tau in the hippocampus and medial prefrontal cortex. This decrease in phosphorylation may be regulated by a lower activity of glycogen synthase kinase-3 beta (GSK3β) and CKD5 related to Tau hyperphosphorylation and neurofibrillary tangle formation. Studies have shown that NAC destroys disulfide bridges by disrupting the self-acetylation of the catalytic domain in microtubules, thereby inhibiting the formation of tau filaments [12,46,49].

Tau hyperphosphorylation is also related to an increase in MAPK activity. Thus, NAC directly inhibits this phenomenon [69]. In this regard, it has been reported that NAC supplementation inhibited MAPK signal transduction by attenuating Tau phosphorylation. By its antioxidant character, NAC mitigates the phosphorylation of ERK, p38 kinase, and JNK (MAPK), thereby increasing the survival of primary rat hippocampal neurons and exerting a neuroprotective effect against oxidative damage, which is a characteristic of AD [73]. In the same vein, by decreasing oxidation levels, NAC inhibited the transcription of factors associated with the expressions of NF-κB an activator protein 1 (AP-1) and attenuated MAPK activity [50].

## 8. Effect of NAC as an Attenuator in the Generation of AGEs-RAGE in AD

Treatment with NAC slightly reduced plasma glucose levels in diabetic rats, and it was speculated that, by decreasing oxidative stress levels in muscle, pancreatic, and adipose tissues, NAC produced the benefit of reduction in plasma glucose levels [57]. In a murine model of hyperglycemia, NAC supplementation reduced the levels of RAGE, ROS, and inflammatory markers such as IL-1β, TNF-α, IL-10, and IL-4, and also inhibited NOX [74].

The mechanism of action involved in the interaction between AGEs-RAGE and NAC is not yet fully described. The administration of NAC inhibited NF-κB activity through loss of its phosphorylation, resulting in reduced expression of RAGE, which prevented the implicit damage induced by AGEs in AD patients [75,76,77,78]. Furthermore, it is possible that NAC presents molecular binding affinity for RAGE, which confers it with therapeutic capacity. Other theories indicate that NAC, by decreasing ROS levels, lowers the expression of RAGE. On the other hand, it has been indicated that NAC reduced the expression of the ligand high mobility group box 1 (HMGB1) and the formation of AGEs, both of which affect and suppress the expression of RAGE [75].

## 9. Conclusions

In Alzheimer’s disease (AD), ROS/RNS levels are accentuated, with a drop in the antioxidant response, leading to an environment conducive for the development of neuroinflammation and progression of the pathology. Due to its antioxidant characteristics, NAC supplementation down-regulates the expressions of NOX-4, Coc, COX-1, and COX-2 and decreases MAO activity. Moreover, NAC reestablishes mitochondrial dysfunction by attenuating ROS/RNS levels and restoring enzyme complexes involved in oxidative phosphorylation, thereby preserving an adequate rate of mitochondrial respiration. Somehow, the return to a functional balance between oxidant and antioxidant species favors the inhibition of pathways such as MAPK, JNK, and p38MAPK. This attenuates the oligomerization of Aβ peptides as well as the hyperphosphorylation of Tau and the formation of neurofibrillary tangles. In addition, NAC supplementation decreases the formation of AGEs and reduces their interactions with their receptors. When the above phenomena are added together, a synergy is observed, which may explain the improvement in the cognitive performance tests applied in clinical trials and in the models described in this study. At the current state of knowledge, as presented in this article, although there are logical premises to suggest that NAC may be beneficial, there is still a lack of conclusive data to suggest such an effect.

A limitation of this work is the small number of clinical trials reported in the literature in AD patients given NAC treatment. Even the murine and in vitro models described are too few in number for the benefits of NAC supplementation in this pathology. Therefore, there is a gap in the opportunity for study in this regard.

## Figures and Tables

**Figure 1 brainsci-15-00164-f001:**
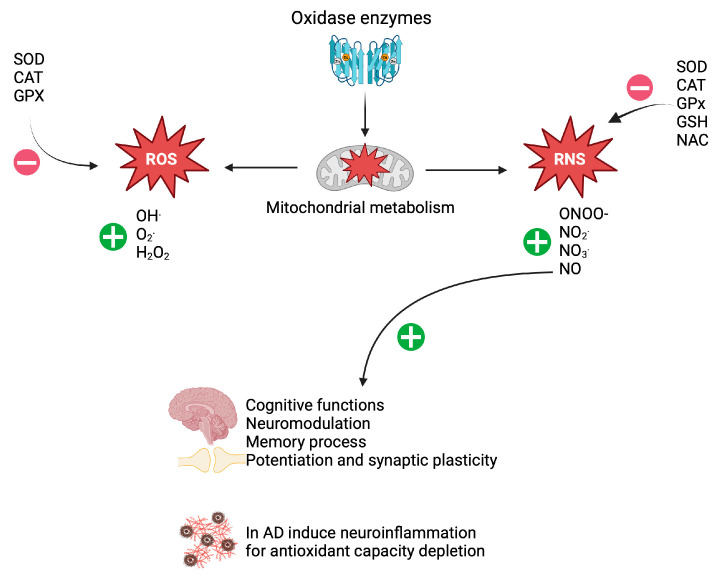
Mechanisms through which antioxidants neutralize ROS/RNS. +, stimulates; −, decreases. Created in BioRender. Bitzer Quintero, O. (2025) https://BioRender.com/k70r471 (accessed on 6 January 2025).

**Figure 2 brainsci-15-00164-f002:**
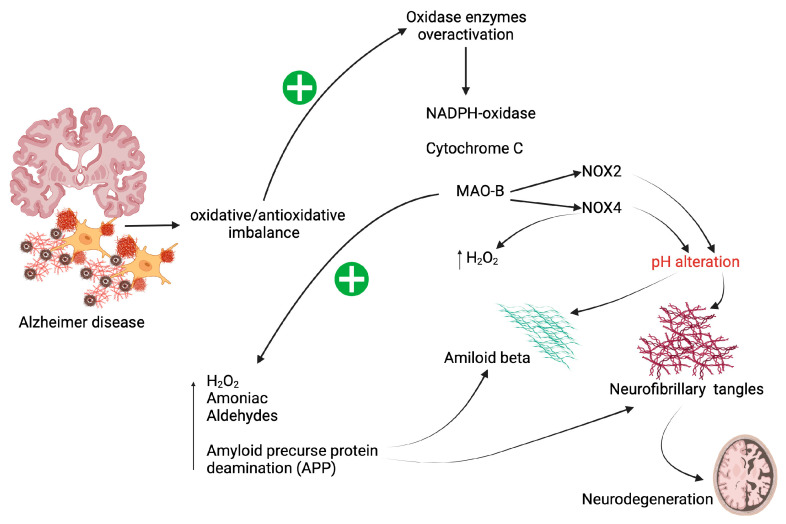
Factors that exacerbate oxidative damage in AD. Created in BioRender. +, stimulates; ↑, increases. Created in BioRender. Bitzer Quintero, O. (2025) https://BioRender.com/w04l325 (accessed on 6 January 2025).

**Figure 3 brainsci-15-00164-f003:**
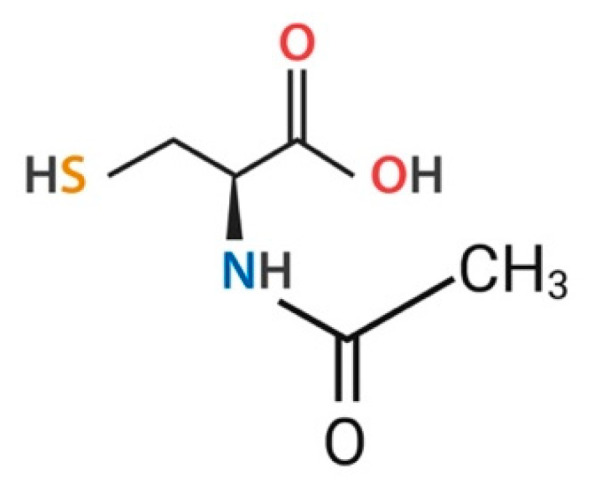
The chemical structure of NAC is formed by the union of cysteine with an acetyl group. Created in BioRender. Torres, D. (2024) https://BioRender.com/k37c819 (accessed on 6 January 2025).

**Figure 4 brainsci-15-00164-f004:**
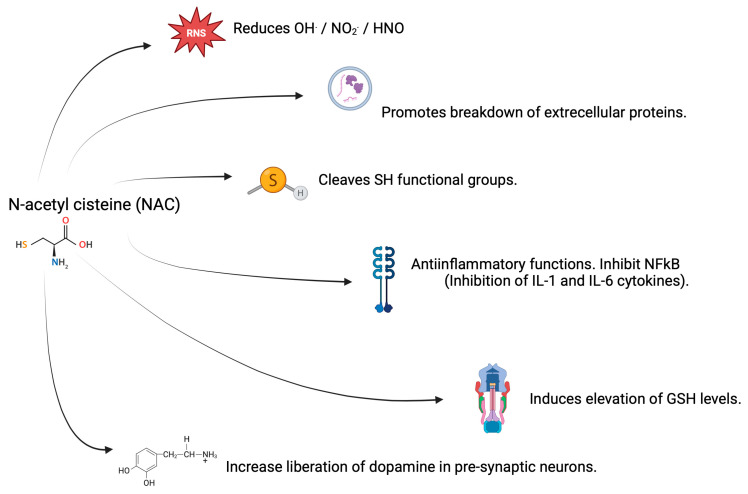
Mechanisms of action of NAC. Created in BioRender. Bitzer Quintero, O. (2025) https://BioRender.com/q81t228 (accessed on 6 January 2025).

**Table 1 brainsci-15-00164-t001:** Effect of NAC supplementation on inflammatory, apoptotic oxidative status and cognitive impairment.

Factors to Evaluate with NAC Supplementation	Study Population Characteristics	NAC Supplementation Characteristics	NAC Supplementation Response	Cognitive Response to Supplementation with NAC	References
Clinical Trials					
Evaluate NAC in patients diagnosed with probable AD.	(1) Patients with probable diagnosis AD + NAC; (n = 23) (2) Patients with probable AD diagnosis + placebo; (n = 20) Patients were selected with MMSE scores ranging from 12 to 26. Patients were free of other dementias and substance abuse.	Group 1 received supplementation with NAC 50 mg/kg per day. Group 2 received placebo. Supplementation with NAC or placebo was supplemented for 6 months.	The oxidative status of patients in both groups before and after 6 months and 6 weeks was evaluated by GPx activity, GSH and thiobarbiturate-reactive. When comparing the NAC-supplemented group with placebo, no differences in marker levels were observed.	It was evaluated by psychometric and cognitive response tests (week 12 and 24) using the MMSE, BNT and WMS tests. No differences are reported in the assessment of the MMSE test between group 1 and group 2 at 24 weeks after the intervention, although a positive trend is observed in the MMSE scale for the group supplemented with NAC.	[39]
To evaluate the effect of nutraceutical formulation (NF) in institutionalized patients with moderate to late-stage AD on their cognitive response	(1) Patients diagnosed with probable AD + NF; (n = 6) (2) Patients diagnosed with probable AD + placebo; (n = 6) Cohort study of institutionalized patients with moderate to late-stage Alzheimer’s disease with a mean MMSE of 11.9 ± 2.5.	Group 1 received NF with 400 μg folic acid, 6 μg vitamin B12, 30 UI alpha-tocopherol, 400 mg S-adenosylcysteine, 600 mg NAC and 500 mg acetyl-L-carnitine in 2 tablets daily. NF and placebo were co-administered for 9 months.	No other parameters were assessed.	It has been evaluated by DRS-2, CLOX-1 and NPI. Patients treated with NF showed a better clinical response on the CLOX-1 and DRS-2 scales compared to the placebo group. They also show 30% improvement in NPI scores, which measure behavioral impairment in AD patients.	[41]
To evaluate the effect of a nutraceutical formulation (NF) on cognitive performance in patients with a presumptive diagnosis of moderate to late-stage AD.	(1) Patients diagnosed with probable AD + NF; (n = 62) (2) Patients diagnosed with probable AD + placebo; (n = 44) Double-blind, Phase II study with a mean score of 22.2 ± 5.1 on the MMSE test	Group 1 received NF with 400 μg folic acid, 6 μg vitamin B12, 30 UI alpha-tocopherol, 400 mg S-adenosylcysteine, 600 mg NAC and 500 mg acetyl-L-carnitine in 2 tablets daily. NF and placebo were administered for 3 to 6 months.	No other parameters were assessed.	CLOX-1 and DRS AEMSS scores were better in the NF-supplemented patients. Additionally, caregivers reported significant improvements on neuropsychiatric tests for supplemented participants.	[42]
Alzheimer’s Animal Models					
Evaluation of cognitive response, hippocampal neuronal loss, oxidative stress markers and antioxidant response	(1) Male Kunming mice with AD model; (n = 8) (2) Male Kunming mice with AD model + NAC50; (n = 8) (3)Male Kunming mice with AD model + NAC100; (n = 9) (4) Male Kunming mice with AD model + NAC2000; (n = 9) (5) Control normal mice (n = 10). Hippocampal stereotaxic surgery was used to induce the AD model by administering 4 μL of Aβ.	Group 1 and group 5 did not receive any supplement. Group 2 received 50 mg/kg i.p. NAC. Seven days prior to Aβ peptide infusion. Group 3 received 100 mg/kg i.p. NAC. Seven days before Aβ peptide infusion. Group 3 received 200 mg/kg i.p. NAC. Seven days before Aβ peptide infusion.	Histological evaluation showed less neuronal loss in the hippocampus in group 2, 3 and 4 rats compared to group 1. Group 1 * shows a high level of MDA and a low level of GSH. For MDA and GSH, groups 2 to 4 show opposite effects. The AChE enzyme activity is increased in group 1, while the AChE values in groups 2 to 4 are similar to those of the control group.	Learning and memory capacity were evaluated. Compared to group 1 *, groups 2 to 4 showed higher retention and shorter latency times in the water maze.	[44]
Electrophysiological evaluation of hippocampus and cognitive response	(1) Male Wistar rats with AD model; (n = 7) (2) Male Wistar rats with AD model + NAC200 day 1; (n = 7) (3) Male Wistar rats with AD model + NAC200 day 14; (n = 7) (4) Control male Wistar rats; (n = 7). Hippocampal stereotaxic surgery was used to induce the AD model by administering 6 μL of Aβ.	No supplementation was given to groups 1 and 4. Group 2 received 200 mg/kg per day of NAC by i.p. injection. Treatment started on day 1 of AD model induction and continued until day 14. Group 3 received 200 mg/kg per day of NAC by i.p. injection. Treatment started on day 14 of AD model induction and continued until day 28.	Groups 2 and 3 show improvement in electrophysiological studies of hippocampal long-term potentiation. Group 1 * is weaker. NAC was able to restore synaptic plasticity in the hippocampus in groups 2 and 3 in comparison to group 1.	Passive avoidance performance was assessed. Compared to group 1 *, groups 2 and 3 show a decrease in the latency to enter the dark compartment. NAC has been shown to slow the progression of cognitive impairment.	[45]
Neuronal loss, tau expression, and cognitive response	(1) Male albino Wistar rats with AD model; (n = 12) (2) Male albino Wistar rats with AD model + NAC50; (n = 12) (3) Male albino Wistar rats with AD model + NAC100; (n = 12) (4) Control male albino Wistar rats; (n = 12) The AD model was induced with colchicine (15 μg into ventricle stereotaxically).	Group 1 and Group 4 were not re-supplemented. Group 2 received 50 mg/kg per day of NAC by i.p. injection. Group 3 received 100 mg/kg per day of NAC by i.p. injection. The total duration of the treatment is 26 days.	Compared to group 1, groups 3 and 4 showed an increase * in the number of neurons in the hippocampus. Between groups 3 and 4 and the control there were no differences. In groups 3 and 4, the presence of tau-positive cells in the hippocampus is less pronounced than in group 1 *.	It was evaluated by the performance of the passive avoidance. Groups 2 and 3 show increased latency to enter the dark compared to Group 1 *. NAC was effective in the reversal of colchicine-induced memory impairment.	[46]
Assessment of cognitive response and GSH/GSSG levels.	(1) Male Sprague-Dawley rats with AD model; (n = 5) (2) Male Sprague-Dawley rats with AD model + NAC; (n = 5) (3) Male Sprague-Dawley control rats + NAC; (n = 5) (4) Male Sprague-Dawley control rats; (n = 5) The AD model was induced with Aβ1-42 peptide (100 μM in phosphate saline) by stereotactic hippocampal surgery in the CA3 region.	No supplementation was given to groups 1 and 4. Groups 2 and 3 received 200 mg/kg NAC orally supplemented for 21 days.	Group 1 has low levels of GSH and GSH/GSSG ratio in comparison to group 4. In contrast to group 1 *, GSH levels and the GSH/GSSG ratio are elevated in groups 2 and 3. Endogenous antioxidant levels in the rat hippocampus are preserved by NAC supplementation. NAC decreased ERK1/2 phosphorylation and increased RyR2 protein levels, which regulate the release of Ca^2+^ ions involved in memory processes.	Spatial memory tests were used to assess it. Group 2 has an improvement in memory in comparison to group 1 *. However, there are no significant differences when Group 2 is contrasted with Group 3. It is reported that supplementing with NAC can reverse the effects of Aβ peptide 1-42.	[47]
Evaluation by Aβ aggregation, cognitive decline, effect on astrocytes and microglia.	(1) Male albino Wistar rats with AD model; (n = 12) (2) Male albino Wistar rats with AD model + NAC50; (n = 12) (3) Male Wistar albino rats with AD model + NAC100; (n = 12) (4) Control male albino Wistar rats; (n = 12) The AD model was induced with colchicine (15 μg) stereotaxically into the ventricle of the brain.	No supplementation was given to groups 1 and 4. Group 2 received 50 mg/kg per day of NAC by i.p. injection. Group 3 received 100 mg/kg per day of NAC stereotactically into the ventricle of the brain. The total duration of the treatment was 26 days.	The number of Aβ aggregates in the hippocampus was lower in groups 2 and 3 with NAC supplementation compared to group 1 *, except for CA1, where the amount of Aβ aggregates is higher even in the supplementation groups There was no significant difference in the expression of reactive astrocytes in the hippocampus in groups 1 to 3. In contrast, NAC supplementation (groups 2 and 3) reduced hippocampal microglial activation, particularly in CA1 and CA4, compared to group 1 *.	It was evaluated using shock avoidance/day and retention test scores. NAC supplementation shows an increase in mean shock avoidance/day during learning and during the memory retention test compared to rats in the AD model group.	[48]
Assessment of cognitive responsiveness, markers of inflammation, antioxidant activity, oxidative stress, Aβ levels, and tau protein phosphorylation.	(1) Male Wistar rats with AD model; (n = 6) (2) Male Wistar rats with AD model + NAC + RUT; (n = 6) (3) Male Wistar rats + NAC + RUT; (n = 6) (4) Male Wistar control rats; (n = 6) Scopolamine (2 mg/kg i.p.) was used for 10 weeks to induce the AD model.	No supplementation was given to groups 1 and 4. A daily oral supplement of 200 mg NAC + 75 mg RUT/kg body weight was administered to groups 2 and 3 for 10 weeks.	The following results were obtained with NAC + RUT supplementation: (1) MDA levels decreased in Group 2 compared to Group 1 *. (2) Increased levels of GSH, SOD, CAT GST and GPx in rats of group 2 compared to rats of group 1 *. (3) Reduction of TNF-α and IL-6 in group 2 compared to group 1 *. (4) Reduction of Aβ1-40 and Aβ1-42 aggregates by 19.21% to 21.08% in the hippocampus of group 2 rats in comparison to group 1 *. (5) 69.84% increase in Nrf2 expression, 53.01% decrease in NOX-2 expression, and 47.96% decrease in BACE1 expression in supplemented rats. (6) A 19.8% decrease in tau phosphorylation in the rats of group 2 as compared to group 1 *.	The supplementation with NAC + RUT resulted in a higher performance in the Morris water maze test as compared to the group 1 *.	[49]
Assessment of cognitive response, neurogenesis, Aβ aggregation, tau protein expression, levels of oxidative markers, and total endogenous antioxidants.	(1) Male Wistar rats with AD model; (n = 18) (2) Male Wistar rats with AD model + amide NAC; (n = 18) (3) Male Wistar rats pretreated with amide NAC + AD model; (n = 18) (4) Control male Wistar rats; (n = 18). The AD model was induced by infusion of 5 μL of Aβ1-42 peptide Aβ (5 μg/5 μL saline) into the hippocampus by stereotaxic surgery.	No supplementation was given to groups 1 and 4. Group 2 received 75 mg/kg per day i.p. NAC amide from day 1 of AD model induction until day 7. Group 3 received 75 mg/kg per day NAC amide i.p. for 7 days, after the seventh day was treated with peptide Aβ1-42, then continued with NAC amide i.p. for 7 more days.	Neurogenesis was evaluated by expression of DCX protein, NAC supplementation increased the levels of this protein in the hippocampus and dentate gyrus, showing increased neuronal proliferation. Similarly, NAC administration reduces the amount of Aβ aggregation in the hippocampus, although it is not clear whether this is due to suppression of misfolding or its clearance. In the hippocampus and medial prefrontal cortex, NAC also reduces tau hyperphosphorylation and MDA levels. On the other hand, NAC increased the levels of GSH and total antioxidants in the hippocampus and medial prefrontal cortex compared to group 1 *.	Spatial learning, memory, and passive avoidance were the outcome measures. A significant decrease in escape latency, distance traveled to the platform, and time spent in the light compartment was observed in animals receiving NAC amide. On the other hand, NAC-supplemented groups increased time on retention tests compared with group 1 *.	[12]
Assessment of cognitive response, the number of hippocampal neurons in CA1, and markers of oxidative stress.	(1) Male Wistar rats with AD model; (n = 8) (2) Male Wistar rats with AD model + NAC amide; (n = 8) (3) Control male Wistar rats; (n = 8) The AD model was induced by infusion of 8 μL of Aβ peptide (8 μg/20 μL saline) into the hippocampus by stereotaxic surgery.	No supplementation was given to groups 1 and 3. Group 2 received 100 mg/kg NAC i.p. daily for 11 weeks after the infusion of Aβ peptide.	There were no differences between groups 1 and 2 in the death of CA1 neurons in the hippocampus. Similarly, there were no significant differences in MDA, SOD, CAT, and FRAP levels between groups 1 and 2.	It was assessed by passive avoidance performance. Group 2 with NAC does not show any difference in the retention time of the memory compared to group 1.	[50]
In Vitro Studies					
Assessing GSH Levels, Aβ Aggregation and Tau Phosphorylation in SHSy5 Neuroblastoma Cells	(1) SHSy5y + 50 μM H_2_O_2_ + 30 mM NAC/or without + 10 μM Aβ25-35 (2) SHSy5y + 50 μM H_2_O_2_ + 30 mM NAC/or without + 10 μM Aβ35-25 (3) SHSy5y + 50 μM H_2_O_2_ + 30 mM NAC/or without + 10 μM Aβ41-42 (4) SHSy5y + 1.2mJ UV (30s) + 30 mM NAC/or without + 10 μM Aβ25-35 (5) SHSy5y + 1.2 mJ UV (30 s)+ 30 mM NAC/or without + 10 μM Aβ35-25 (6) SHSy5y + 1.2 mJ UV (30 s)+ 30 mM NAC/or without + 10 μM Aβ41-42 (7) SHSy5y + 50 μM H_2_O_2_ + 30 mM NAC/or without + 10 μM Aβ25-35 + 200 μM GSH (8) SHSy5y + 50 μM H_2_O_2_ + 30 mM NAC/or without + 10 μM Aβ35-25 + 200 μM GSH (9) SHSy5y + 50 μM H_2_O_2_ + 30 mM NAC/or without + 10 μM Aβ41-42 + 200 μM GSH (10) SHSy5y + 1.2 mJ UV (30 s) + 30 mM NAC/or without + 10 μM Aβ25-35 + 200 μM GSH (11) SHSy5y + 1.2 mJ UV (30 s)+ 30 mM NAC/or without + 10 μM Aβ35-25 + 200 μM GSH (12) SHSy5y + 1.2 mJ UV (30 s)+ 30 mM NAC/or without + 10 μM Aβ41-42 + 200 μM GSH (13) SHSy5y + 50 μM H_2_O_2_ + 30 mM NAC/or without + 10 μM Aβ25-35 + MTT test (14) SHSy5y + 50 μM H_2_O_2_ + 30 mM NAC/or without + 10 μM Aβ35-25 + MTT test (15) SHSy5y + 50 μM H_2_O_2_ + 30 mM NAC/or without + 10 μM Aβ41-42 + MTT test (16) SHSy5y + 1.2 mJ UV (30 s) + 30 mM NAC/or without + 10 μM Aβ25-35 + MTT test (17) SHSy5y + 1.2 mJ UV (30 s) + 30 mM NAC/or without + 10 μM Aβ35-25 + MTT test (18) SHSy5y + 1.2 mJ UV (30 s) + 30 mM NAC/or without + 10 μM Aβ41-42 + MTT test (19) SHSy5y + Solution Buffer (Control) For all groups, 5000 cells/well at 37 °C were used. The groups were exposed to H_2_O_2_ for 30 min. The UV exposure was performed by two bursts of irradiation for 30 s at 1.2 mJ with an interval of 15 min.	All groups except control received 30 mM NAC preincubated for 30 min.	NAC-treated groups show GSH and MTT levels similar to controls despite UV exposure, with the exception of cells that have been treated with H_2_O_2_. A higher release of Aβ41-40 and Aβ41-42 was observed in cells exposed to H_2_O_2_ without NAC. In the group with Aβ25-35 and oxidative damage and treated with NAC, the Aβ41-40 residues are released. UV-exposed cells release Aβ41-40 and Aβ41-42 residues and pre-treated with NAC does not change. Pre-incubation of the cells with NAC stimulates the dephosphorylation of tau in spite of the oxidative damage they have received.	Not applicable	[51]
Oxidative stress markers, mitochondrial dysfunction and apoptotic markers.	(1) AD patients’ fibroblasts (2) Fibroblasts obtained from control subjects	Group 1 was treated with 100 μM of NAC in PBS for 24 to 48 h. Both groups were maintained at 37 °C with 5% CO_2_ in 1X DMEM supplemented with antibiotics, antifungals, and glutamine.	NAC supplementation in Group 1 shows decreased HNE, CML and OH-1 levels. Mitochondrial dysfunction was assessed by inhibiting ferrochelatase using NMP, which NAC supplementation attenuates. Similarly, in fibroblasts from AD patients, NAC attenuated the levels of Bax and caspase 9.	Not applicable	[52]
Tau aggregation and neuritogenesis	(1) Mouse neuroblastoma cells N2a (7000 cells/well) (2) Cells N2a (7000 cells/well) (3) Sequence htau40 (50 μM) with heparin (4:1) for 10 days (4) Sequence htau40 (50 μM) without heparin	Group 1 N2a cells were treated with NAC at varying concentrations of 1 to 15 μM for 24 h at 37 °C, 5% CO_2_. No NAC treatment was given in groups 2 and 4. Group 3 received 5 mM NAC on the third day of incubation with heparin.	Compared to the control group, NAC supplementation reduced htau40 aggregation by 60%. In N2a cells, supplementation with NAC resulted in a significant increase in the number of neurites in this cell line in a dose-dependent manner.	Not applicable	[28]

* *p* < 0.05; AD, Alzheimer’s Disease; NAC, N-acetylcysteine; DCX, doublecortin protein; MDA, malondialdehyde; SOD, superoxide dismutase; CAT, catalase; FRAP, Ferric Reducing Ability of Plasma; GSH, reduced glutathione; GSSG, Glutathione disulfide; AChE, acetylcholinesterase; MMSE, Mini-Mental State Examination; BNT, Boston Naming Test; WMS, Wechsler Memory Scale; DRS-2, Dementia Rating Scale 2; CLOX-1, Clox Drawing Test-1; NPI, Neuropsychiatric Inventory; NF, nutraceutical formulation; AEMSS, age- and education-adjusted; TNF-α, Tumor Necrosis Factor Alpha; IL-6, interleukin-6; RUT, rutin; GST, Glutathione S-Transferase; GPx, Glutation peroxidase; NOX-2, NADPH oxidase-2; BACE1, Beta secretase-1; H_2_O_2_, hydrogen peroxide; UV, ultraviolet radiation; mJ, million Joules; MTT, colorimetric test to determine cell cytotoxicity; DMEM, Dulbecco’s Modified Eagle Medium; HNE, 4-Hydroxynonenal; CML, N-(carboxymethyl)lysine; OH-1, Heme oxygenase-1; NMP, N-methylprotoporphyrine IX; Nrf2, Nuclear factor erythroid 2-related factor 2.

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
