# Peer review of "Effect of N-Acetyl Cysteine as an Adjuvant Treatment in Alzheimer’s Disease"

_brainsci, 2025, doi:10.3390/brainsci15020164_

Round 1

Reviewer 1 Report

Comments and Suggestions for Authors

The authors did a great job of analyzing the available literature on the possible effects of NAC on AD.

In the introduction, the authors discuss the properties and effects of NAC, however:

line 220

The use of NAC in COPD as a mucolytic drug and upper respiratory tract infections is worth mentioning because it does not increase the volume of mucus.

line 223

Adverse effects are indeed listed, but it seems that it should be given a little more attention, focusing on the fact that AD is a chronic disease - for the rest of life, so if we intend to use the drug chronically, we should consider the possible problems associated with it. It is not comparable to a single administration of paracetamol in an overdose or a cold. 

Please look:

Ershad M, Naji A, Patel P, et al. N-Acetylcysteine. [Updated 2024 Feb 29]. In: StatPearls [Internet]. Treasure Island (FL): StatPearls Publishing; 2025 Jan-. Available from: https://www.ncbi.nlm.nih.gov/books/NBK537183/

Tsai, M.-S.; Liou, G.-G.; Liao, J.-W.; Lai, P.-Y.; Yang, D.-J.; Wu, S.-H.; Wang, S.-H. N-acetyl Cysteine Overdose Induced Acute Toxicity and Hepatic Microvesicular Steatosis by Disrupting GSH and Interfering Lipid Metabolisms in Normal Mice. Antioxidants 2024, 13, 832. https://doi.org/10.3390/antiox13070832

There is no analysis of NAC dosage. The table compares different doses of 600mg, 100mg/kg-200mg/kg up to 5000mg/kg. It would be worth discussing what effect the dose has on the expected effect? ​​600mg is a dose of popular cough medicines, so it seems that physicians prescribed this medicine as available on the market. But maybe the differences in the obtained effects result from differences in dosage?

Table 1. Effect of NAC supplementation on inflammatory, apoptotic oxidative status and cognitive impairment, animal models first example. 

[42] - it is on mice not rats, it is not clear what this data is from.

[38] - a case study should not be considered evidence in assessing the usefulness of NAC in the treatment of AD.

[43] - why do the authors use the experience of TBI in the analysis of AD studies? The mechanism is not the same, it cannot be considered equivalent.

[44] - the same - STZ-induced cognitive impaired rats.

In conclusion, the Authors did not answer whether NAC supplementation is beneficial or not in AD. In my opinion, it should be clearly stated that in the current state of knowledge, presented in this work, although there are logical premises that NAC may be beneficial, there is no data indicating such an effect.

Author Response

Thank you very much for your review, we consider your contributions valuable to the improvement of this work. In the attached file we indicate our response for each of your comments. As you can see in the attached document, we also detail the page number and lines where you can check the change made in the article.

Reviewer 2 Report

Comments and Suggestions for Authors

The purpose of the current review was to review the research on NAC supplementation as an adjunct treatment for AD.  This was accomplished through a major literature review containing 9 major sections and extensive table. Briefly the main findings and conclusions were that NAC improves cognitive performance, decreases oxidative stress markers, decreases oxidase enzyme activity, increases antioxidant responses, and decreases a number of inflammatory and apoptotic markers. Furthermore, NAC seems to improve mitochondrial function, lowers AGEs-RAGE formation, attenuates Aβ peptide oligomerization. Accordingly, it seems to decrease of tau phosphorylation, slowdown the development of neurofibrillary tangles, and perhaps even slow the progression of AD.

 The study had several strengths. The study was very well-written, easy to understand, and had few typographical or grammatical errors. The topic of research (AD) is one of the most important ones in science today and non-invasive, affordable, adjunct treatments like NAC are sorely needed in AD.  The authors included limitations, which was an additional strength. The table was helpful and extensive.

Accordingly, I only have as set of minor comments that I feel it would be helpful for the authors to address and improve an already good paper.

1. It may be slightly outside the scope of the review but I am wondering if somewhere in the introduction or conclusions the studies of the Sekhar RV research group (I am not a member of the group) should be mentioned. Those studies combine NAC and Glycine and perhaps a few other compounds and have gotten very impressive results in older adults. It may be out of scope as it combines NAC with other compounds, but perhaps this should be mentioned given the number of studies on the topic and the strong results.

2. Very recently, I came across some research that NAC may damage the gut lining, which could ultimately increase inflammation and cause other problems. I have not had time to look into this issue, but this was disturbing to me as someone who has taken NAC in the past and a proponent of it based on research like mentioned above. I bring this up for two reasons. First, perhaps the authors should have a small section covering the side effects of NAC (if any) just so the reader is aware. Second, perhaps the authors should look into this gut lining issue and mention it if applicable. With any supplement, benefits must always be weighed with side effects so perhaps a short mention in the intro or conclusions should be done. As in the above point, I am not suggesting the authors add these studies to their tables etc just mentioning these issues as background info.

3. There may be a few instances of too many spaces between words in the text such as in the abstract so check this.

4. Table 1 provides a lot of great information, I am just wondering if the formatting can be improved as there are big spaces between some words and sentences etc. Perhaps there is a way to cut down on some of the info a bit also.

Author Response

Thank you very much for your comments, we consider your contributions valuable for the improvement of this work. In the attached file we indicate our response for each of your comments. As you can see in the attached document, we also detail the page number and lines where you can check the change made in the article.

Round 2

Reviewer 1 Report

Comments and Suggestions for Authors

The authors have corrected the work.